# The NEMA Device for Efficient Extraction and Rearing of Entomopathogenic Nematodes

**DOI:** 10.3390/insects16100991

**Published:** 2025-09-23

**Authors:** Camila C. Filgueiras, Jennifer Luna-Ayala, Catherine Anderson, Caroline Kennedy, Denis S. Willett

**Affiliations:** 1Department of Biology, UNC Asheville, One University Heights, Asheville, NC 28804, USA; 2North Carolina Institute for Climate Studies, Department of Applied Ecology, NC State University, 151 Patton Avenue, Asheville, NC 28801, USA

**Keywords:** entomopathogenic nematodes, biological control, sustainable agriculture, nematode extraction, nematode multiplication

## Abstract

Nematodes are tiny worms found in soil that play important roles in farming and the environment. Some types, called entomopathogenic nematodes (EPNs), help naturally control insect pests, reducing the need for chemical pesticides. To study or use these helpful nematodes, scientists need to collect them from soil and raise more in laboratories. Traditional tools for conducting this are often slow, messy, and hard to use outside of a lab. In this study, we designed a new tool called the NEMA Device, made from common plumbing parts, that combines both collecting and raising nematodes into one simple system. We tested the NEMA Device with two different nematode species and found that it recovered more nematodes from soil than standard methods and produced just as many new ones. The device was also easier to use, more portable, and less likely to become contaminated. By making it easier and more reliable to work with beneficial nematodes, the NEMA Device could help more farmers, students, and scientists use these natural pest controllers in both research and agriculture. This could support more sustainable, low-chemical approaches to growing food while making nematode research more accessible around the world.

## 1. Introduction

Nematodes are among the most ubiquitous and diverse organisms on Earth, playing critical roles in nearly all ecosystems and occupying all soil trophic levels [1]. Nematodes are critical contributors to nutrient cycling, soil health, and biological control, positively impacting the natural environments, agriculture, and forestry [2]. These organisms, in addition to performing important ecological services, can parasitize many organisms, including humans. In agroecosystems, they can be both beneficial and harmful, acting as natural biocontrol agents against insect pests or as plant parasites [3].

In agroecosystems, natural nematode biocontrol agents, entomopathogenic nematodes (EPNs), have been effectively used in integrated pest management (IPM) programs to manage a wide range of insect pests [4]. In addition to their practical applications, EPNs serve as an excellent model system for studying host–parasite interactions, symbiosis, and insect pathology [5,6]. Their relatively simple lifecycle, ease of rearing in the laboratory, and well-studied biology make them ideal for both fundamental and applied research [5,6].

Working with EPNs for use in basic research, applied research, and monitoring involves extracting them from soil and rearing them for laboratory or field use. Efficient extraction and rearing of EPNs are essential for their use both as biological control agents and for research purposes.

Nematode extraction from soil can be accomplished using various methodologies [7,8]. The Baermann funnel technique is one of the most widely used methods for extracting mobile nematodes, including EPNs, from soil samples [9]. This method relies on the natural motility of nematodes, allowing them to migrate through a mesh or Kimwipe into water, from where they can be collected [9]. The extraction efficiency of Baermann funnels can vary significantly, depending on soil type, moisture content, the condition of the nematodes, and the construction of the Baermann funnel itself [10,11,12]. Despite this variation, the Baermann funnel extraction efficiencies are mostly around 15% [11].

Recent research has explored alternative devices and modified protocols to address the limitations of conventional Baermann funnels and White traps. For example, modular extraction columns and scaled multi-funnel systems have been developed to improve throughput and standardization [7,12], while innovations such as field-portable extraction kits and automated counting platforms have enhanced mobility and quantitation in remote or resource-limited settings [10,13]. Other designs incorporate interchangeable mesh sizes or adjustable water flow to accommodate different soil textures and nematode sizes [8].

Once extracted, EPNs are commonly reared in insect hosts to increase their populations for subsequent use [9]. Infected insect larvae are typically placed on White traps, a technique that facilitates the collection of EPN infective juveniles as they emerge from the insect host cadaver [9,14]. White traps consist of a Petri dish, or similar container, with a moist filter paper that allows the infective juveniles to migrate from the insect cadaver into a water reservoir, where they can be collected [9].

While these conventional techniques have proven effective, a number of efforts have been made to improve their scale, throughput, and efficiency. These efforts range from slight modifications of traditional approaches using different materials to scaling multiple devices in parallel [12,15].

Despite these developments, opportunities remain, particularly in developing devices that can be used in both field and laboratory environments, where portability, scalability, efficiency, and high throughput are essential. To explore opportunities to improve these characteristics, we compared the performance of traditional methods for EPN extraction and rearing with that of a single, two-in-one, NEMA Device that accomplishes both in a single unit. We hypothesized that, in addition to ease of use, the NEMA Device would yield higher EPN recovery from soil and comparable rearing efficiency to traditional methods.

## 2. Materials and Methods

To evaluate the NEMA Device, we used two species of entomopathogenic nematodes (one larger, one smaller) in two different assays: rearing and extraction. Rearing assays were compared with traditional White trap methods and extraction methods compared with traditional Baermann methods.

### 2.1. Device Construction

Traditional Baermann funnels were constructed from a plastic funnel connected to a 50 mL conical centrifuge tube (Figure 1A—left). Soil samples (50 g) were placed in large Kimwipes (Kimtech Science, Iselin, NJ, USA) and inserted into the funnel. The funnel was then filled with distilled deionized water to reach the sample and left for 24 h, during which time the nematodes descended into the conical tube.

Traditional White traps (Figure 1B—left) were constructed out of nested Petri dishes with the inner dish holding *Galleria mellonella* hosts on a moistened piece of filter paper and the outer water-filled dish serving as a collection receptacle.

The NEMA Device was constructed with three nested PVC components, combining elements of the Baermann funnel and White trap method (Figure 1). The initial PVC component (depicted by the asterisks in Figure 1) is constructed from a round white PVC 7.6 cm diameter snap-in drain (Oatey, Cleveland, OH, USA) with ridges cut into the base to allow free fluid flow. Because of the grid bottom, this component serves a dual function, both as a retainer for soil samples when the NEMA Device is used for soil extraction (Figure 1A) and, when flipped, as a platform for emergence when the NEMA Device is used for rearing (Figure 1B).

The second PVC component, in which the previously described component is placed, is a 10.2 cm diameter white PVC cap (Oatey) with a 1.9 cm diameter hole drilled through the base center. This hole is threaded to attach a brass barbed coupler garden hose adapter fitting (1.9 cm, American Imaginations, Brampton, ON, Canada) connected to a 0.64 cm internal diameter clear PVC vinyl tube (EZ-Flo, Ontario, CA, USA). This tube is fitted with a barbed on/off irrigation valve (RainDrip, Simi Valley, CA, USA), which controls the release of nematodes into collection tubes.

The third PVC component is a 10.2 cm diameter PVC coupler (Oatey), which serves as a holding base for the NEMA Device and has a 2 cm diameter hole drilled in the side through which the tubing is threaded for collection.

### 2.2. Nematode Rearing

Stock cultures of entomopathogenic nematodes were maintained at the Natural Enemy Management and Applications (NEMA) Laboratory, University of North Carolina at Asheville, and reared for this experiment following established methodologies [16]. Briefly, stock cultures were kept in tissue culture flasks at concentrations not exceeding 2500 infective juveniles (IJs) per mL to ensure long-term survival of the population. To avoid potential reduction of activity and survivorship of older EPN populations, only recently emerged (less than two weeks old), highly viable IJs were used for these experiments. To rear new IJs from the stock cultures, *G. mellonella* larvae were inoculated with a known concentration of EPN IJs of a single species. When host larvae became discolored due to infection, they were transferred to a White trap where EPNs began emerging after 2–3 days [14]. The newly emerged IJs moved into the deionized water of the Petri dish where they were collected by pouring the liquid into tissue culture flasks. Collected IJs were then used in the subsequent experiments described below.

Two different nematode species were used in these bioassays: *Heterorhabditis bacteriophora* Poinar, 1976 and *Steinernema khuongi* Stock 2017. Both species are found naturally in soils of the Eastern United States [17,18,19]. *Heterorhabditis bacteriophora* is commonly used as a biological control agent for a wide range of insects while *S. khuongi* is under consideration for its potential use as a highly specific control agent for citrus weevil pests [20,21]. *Heterorhabditis bacteriophora* is a smaller entomopathogenic nematode (about 0.5 mm long), and *S. khuongi* is a larger entomopathogenic nematode (about 1 mm long).

### 2.3. Extraction Assay

To evaluate and compare extraction efficiencies, known amounts of EPN infective juveniles were added to 50 mL samples of autoclaved sand, placed in Kimwipes, then in the traditional Baermann funnels or NEMA Device (configured for extraction as in Figure 1A), and collected after 24 h. Specifically, 3 mL of *H. bacteriophora* stock suspension at a known concentration ranging from 2100 to 2400 IJs/mL were added to 50 mL (80 g) of autoclaved sand. For *S. khuongi*, 4 mL of a 2800 IJs/mL stock suspension were added to 50 mL (80 g) of autoclaved sand. The inoculated sand was placed in large Kimwipes (Kimtech Science) prior to being added to the respective devices. Devices were filled with 100 mL of water. Extraction experiments were conducted twice with ten replications.

Nematodes were collected after 24 h. For the Baermann funnels, this involved the removal of the conical centrifuge tube. For the NEMA Device, the valve was opened to allow collection into a 50 mL conical tube. Collected nematodes were then counted under a dissecting scope.

### 2.4. Rearing Assay

To evaluate and compare rearing efficiencies, single *G. mellonella* larvae were surface-sterilized in a 10% bleach solution and then placed in 60 mm Petri dishes on top of Whatmann Type I filter paper. Larvae were then inoculated with 500 μL of water containing 300 infective juveniles of either *H. bacteriophora* or *S. khuongi*. Petri dishes containing inoculated *G. mellonella* larvae were then added to larger Petri dishes (100 mm) for traditional White trap assays or to the NEMA Device. Deionized water was then used to fill the respective receptacles up to a quarter up the wall of the small Petri dish. Eight replications of each combination of EPN species and assay type were conducted on two separate occasions (time-separated cohorts). Rearing assays were conducted at 23 °C ± 2 °C.

After setup, each assay type was monitored until emergence of infective juveniles from the inoculated *G. mellonella* (about 5–7 days). Emerging IJs were collected from the respective assays into 50 mL conical centrifuge tubes until counting using the Smart Soil Organism Detector (SOD) device [13]. Briefly, the Smart SOD is a custom large bore flow cytometer (COPAS VISION 500, Union Biometrica, Holliston, MA, USA) equipped with imaging and sensing capabilities that allow for high-throughput quantitation and identification of nematodes. Supervised machine learning models (support vector machines) were trained on known specimens and were then used to identify and count nematodes passing through the instrument.

### 2.5. Statistical Analysis

All data were collected into tabular CSV files and then analyzed using the R Statistical Programming Language (version 4.3.3) and the RStudio IDE (version 2024.04.1+748) [22,23]. The following packages facilitated analysis: *tidyverse* [24], *here* [25], *emmeans* [26], *car* [27], *janitor* [28], *tidymodels* [29], *ggsignif* [30], *gghalves* [31], *themis* [32], *foreach* [33], *doparallel* [34], *dorng* [35], *vroom* [36], and *finetune* [37]. Full datasets and code used to produce this analysis are available upon request.

Extraction assay results were evaluated using the recovery rate calculated as the ratio between nematodes recovered and the number of nematodes inoculated. Recovery rate was modeled using linear regression and analysis of variance after considerations of assumptions of normality and homoscedasticity (as evaluated with Shapiro–Wilk’s and Levene’s tests). Post hoc comparisons between assay types (Baermann funnel and NEMA Device) were conducted using the Tukey Method. Confidence intervals of differences between assay types were estimated using non-parametric bootstrapping (1000 replications) and differences between species evaluated with permutation (1000 permutations) tests when data were non-normal.

Rearing assays were evaluated based on total counts of nematodes collected from each assay. Total counts of nematodes were derived from the Smart SOD device output. Differences in rearing outcomes of the two different assays (White traps and NEMA Device) were assessed using permutation tests (1000 permutations). All test results were evaluated for significance at α = 0.05.

## 3. Results

### 3.1. Extraction Assay

Recovery rate from extraction assays was significantly different between the two funnel types. Funnel type significantly explained 29% of the observed variance in recovery rate (F = 10.8, df = 1, p=0.003). The NEMA Device had significantly and consistently higher recovery rates compared with the traditional Baermann funnels (t=3.3, df = 27, p=0.003; Figure 2). The NEMA Device had a 5% (95% confidence interval [CI]: 0.1%, 10%) higher recovery rate for *H. bacteriophora*, and 7% (95% CI: 5%, 9%) higher recovery rate for *S. khuongi* (Figure 3).

### 3.2. Rearing Assay

There were no significant differences between the White trap and the NEMA Device in their output of infective juveniles (p>0.05; Figure 4).

## 4. Discussion

These results suggest that the NEMA Device may demonstrate advantages over traditional methods of extraction and rearing of entomopathogenic nematodes. Extraction using the NEMA Device had consistently higher efficiency across two different genera of different sizes. This increase in efficiency could be due to a number of factors, including improved control over the collection process, the isolated controlled environment within the device, or the shape of the funnel.

Rearing using the NEMA Device was comparable with traditional, high output White trap methods. Although output was comparable on a per-host basis, anecdotally, insect hosts in the NEMA Device had less contamination than traditional Petri dish-based White trap methods where adjusting moisture and adding collection media necessitate opening the device more frequently. The closed nature of the NEMA Device maintained moisture while the larger volume of collection water enabled maintaining an environment where contamination was less likely to occur.

Beyond efficiency, the NEMA Device offers several practical benefits for users. First, the structure of the NEMA Device is stand-alone. It does not require special stands or mounting devices and can be placed on tabletops. A second key advantage is its portability. The use of lightweight PVC components makes the device easy to transport and robust, which is particularly valuable for field studies where researchers need to conduct extractions on-site. The modular design also allows the device to be assembled and disassembled quickly, adding to its convenience for field use. The NEMA Device also has improved user control features. The inline valve allows for precise management of water flow during the collection process, making it easier to collect nematodes without disturbing the sample.

An additional key benefit is the versatility of the NEMA Device. Unlike traditional methods that require different setups for extraction and rearing, the NEMA Device combines both functions into a single unit. This reduces the need for multiple pieces of equipment and simplifies the workflow for researchers who need to switch between extraction and rearing tasks. The ability to use the same device for both processes reduces space needed for running these procedures and reduces the overall cost of equipment.

The advantages of the NEMA Device, including improved extraction efficiency in sand, portability, and versatility, indicate its potential value for both laboratory and field applications. However, beyond the practical benefits, this device has the potential to contribute to the continuing work with both entomopathogenic and other nematodes [8,15,38]. The increased efficiency of the NEMA Device suggests that it may help overcome some of the existing limitations associated with traditional extraction and rearing methods, particularly those related to the variability and lower recovery rates of nematodes from soil [11].

The context provided in the introduction highlighted the critical roles that nematodes play in soil health, nutrient cycling, and biological control, as well as the importance of effective extraction and rearing methods for research and integrated pest management (IPM) programs [2,3]. By improving the efficiency, portability, and ease of use of EPN extraction and rearing, the NEMA Device has the potential to lower operational barriers to incorporating EPNs into IPM strategies. Its low-cost, modular design supports scalability and deployment in both laboratory and field settings, including regions with limited infrastructure. This versatility could facilitate broader adoption of EPN-based biological control by growers, extension personnel, and researchers, enabling more timely and reliable production of infective juveniles for field applications. In this way, the NEMA Device not only advances methodological capacity in nematode research but also enhances the practical feasibility of integrating EPNs into sustainable pest management programs.

The development of the NEMA Device addresses several of these needs by providing a more consistent and effective method for collecting EPNs. This is particularly relevant given the increasing emphasis on sustainable agricultural practices that rely on biological control agents rather than chemical pesticides [20,39,40,41]. By improving the efficiency of extraction in sand and simplifying the process, the NEMA Device could make it easier for researchers to utilize EPNs in IPM programs.

Future research should validate the performance of the NEMA Device under a wider range of operational conditions. In particular, testing in natural, non-sterile soils of varying texture and organic matter content, as well as across different climates, would help assess its robustness for real-world applications. Additional studies could explore scaling the device for higher-throughput extraction, evaluating its compatibility with other nematode species, and integrating it into on-farm IPM workflows. Such work would further clarify its role in facilitating the adoption of EPNs as reliable, field-ready biological control agents.

## 5. Conclusions

While the principal objective of this work was developing a device to facilitate and enhance the ease of use for nematode extraction and rearing, a scalable and portable device like the NEMA Device has the potential to facilitate broader research initiatives and adoption of EPNs as biological control agents, especially in regions where traditional methods are impractical or resources are limited. By making it easier to extract and rear nematodes, this device could help expand the use of EPNs in pest management, contributing to more environmentally friendly and sustainable agricultural practices. Field validation under diverse soil types and environmental conditions, as well as testing at larger operational scales, is recommended to fully establish its utility in applied IPM programs.

## Figures and Tables

**Figure 1 insects-16-00991-f001:**
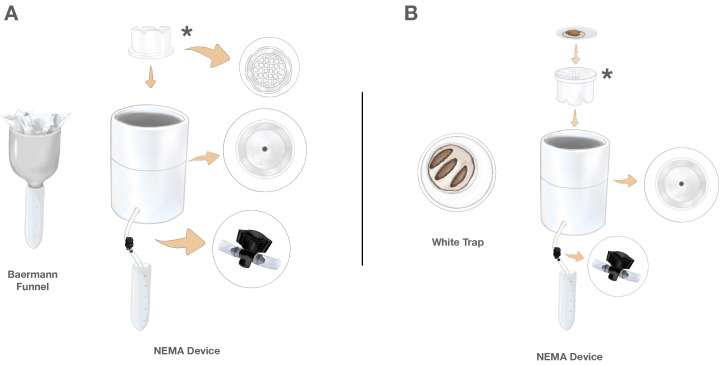
The NEMA Device is a two-in-one device supporting both rearing and extraction. (**A**) **Nematode extraction from soil:** The traditional Baermann funnel is constructed from a plastic funnel connected to a 50 mL conical centrifuge tube. Soil sample is placed in Kimwipes and inserted into the funnel. The NEMA Device is constructed from nested PVC pipes, enabling a soil sample in a Kimwipe to be placed in a PVC cap (asterisk) that allows emerging nematodes to descend into a tube where collection is regulated by an inline valve. (**B**) **Nematode rearing**: The traditional White trap is constructed of nested Petri dishes, with the inner dish holding *Galleria mellonella* hosts and the water-filled outer dish serving as a collection receptacle. The NEMA Device supports nematode rearing through inverting the PVC cap (asterisk), which becomes a platform for a Petri dish containing *G. mellonella* larvae hosts. When infective juveniles (IJs) emerge and enter the surrounding water, they are collected through flexible tubing with an inline valve that leads to a 50 mL centrifuge tube.

**Figure 2 insects-16-00991-f002:**
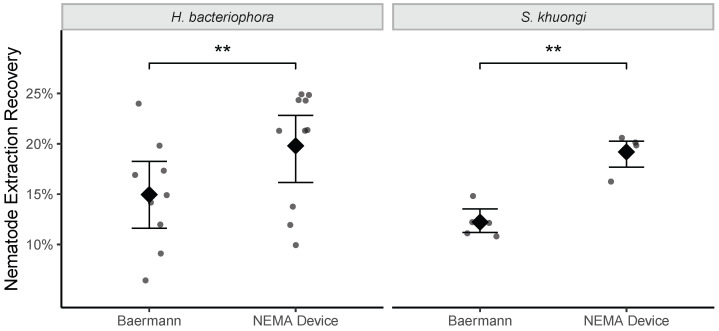
Recovery rates of *H. bacteriophora* and *S. khuongi* entomopathogenic nematode infective juveniles from Baermann funnel or NEMA Device extraction assays. Points represent observed values. Diamond points and error bars denote mean and 95% confidence intervals, respectively. Double asterisks (**) denote significance at p=0.003.

**Figure 3 insects-16-00991-f003:**
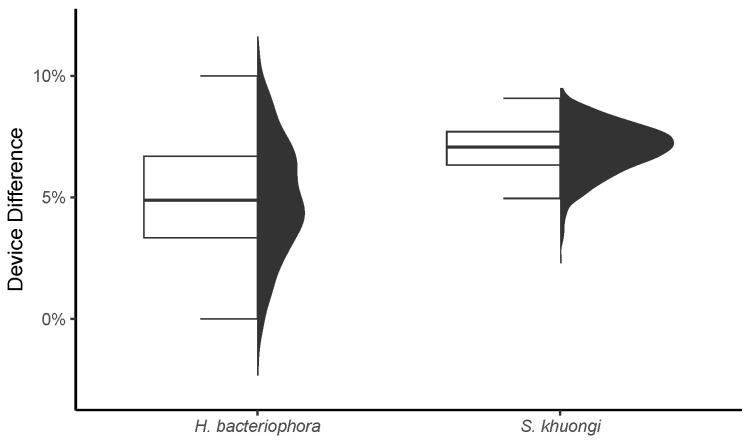
Bootstrapped differences in recovery rate between NEMA Device extraction and traditional Baermann funnels. Left box plots depict the median and interquartile range (IQR). Whiskers denote the IQR ± IQR. Right density distributions reflect distribution differences bootstrapped 1000 times.

**Figure 4 insects-16-00991-f004:**
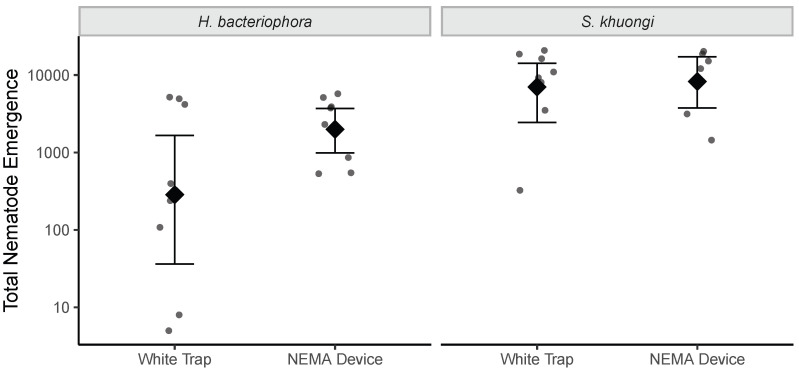
Emergence of *H. bacteriophora* and *S. khuongi* entomopathogenic nematode infective juveniles from *G. mellonella* hosts in NEMA Devices and traditional white traps. Points represent observed values. Diamond points and error bars denote mean and 95% confidence intervals, respectively.

## Data Availability

The original contributions presented in this study are included in the article. Further inquiries can be directed to the corresponding author.

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
