# Peer review of "The NEMA Device for Efficient Extraction and Rearing of Entomopathogenic Nematodes"

_insects, 2025, doi:10.3390/insects16100991_

Round 1
Reviewer 1 Report
Comments and Suggestions for Authors
This manuscript describes the design and evaluation of a novel, low-cost PVC device (NEMA Device) that serves dual purposes: extraction and rearing of entomopathogenic nematodes (EPNs). The device is compared to standard Baermann funnels (for extraction) and White traps (for rearing), using Heterorhabditis bacteriophora and Steinernema khuongi. The topic is timely and relevant for researchers working in applied entomology and nematology. The study is clear in its intent and design, but several important revisions are needed to improve clarity, reproducibility, scientific precision, and consistency.
Introduction
“Nematodes play critical roles in nutrient cycles, provide ecological services, and can be parasites…” (line 25)
Comment: This is a repetition of lines 21–24. “...and nematodes in the soil contribute to nutrient cycling... ” Please revise for conciseness. Redundancy should be avoided.
There is no clearly stated research question or hypothesis.
Recommendation: Add a specific hypothesis, such as (for example): “We hypothesised that the NEMA Device would yield significantly higher EPN recovery from soil and comparable rearing efficiency to traditional methods.” The introduction outlines background well, but lacks a clear problem statement or testable hypothesis.
Materials and Methods
Rearing Assay
“Eight replications... on two separate occasions.” (line 133)
Comment: Specify “occasion”.
Statistical Analysis
“...sumptions of normality and homoscedasticity...” Recommendation: Shapiro-Wilk? Levene? It is recommended that the authors explicitly state which statistical tests were used to assess the assumptions of normality and homoscedasticity.
Recommendation: Indicate how many permutations were used. How many bootstrap replicates? (e.g., 1000?)
The manuscript does not clearly state the significance level (α) used in statistical analyses. Although a threshold of 0.05 is often assumed, I recommend that the authors explicitly indicate the chosen alpha level to ensure transparency and reproducibility.
- Have any corrections been applied in the statistical analysis?
Results
“Funnel type significantly explained 29%...” Comment: What is the full model R² ? This value (29%) seems isolated.
Discussion
“...anecdotally insect hosts in the NEMA Device had less contamination...” (line 182)
Comment: “Anecdotally” is unacceptable in scientific writing. Unless contamination was systematically quantified, this claim should be removed. Replace anecdotal statements with quantified results or omit.
“The closed nature... larger volume of collection water...” (lines 185-186)
Comment: A valuable insight, but speculative without moisture or contamination data.
Comment: Although the Introduction briefly mentions the application of entomopathogenic nematodes (EPNs) in pest control, the Discussion section does not sufficiently elaborate on how the NEMA Device contributes to integrated pest management (IPM) frameworks. This represents a missed opportunity to situate the findings within a broader context of sustainable pest management.
Recommendation: I suggest the authors expand the Discussion by explicitly addressing the potential implications of the NEMA Device for IPM strategies. In particular, the device's scalability, cost-effectiveness, portability, and ease of use could be discussed in terms of facilitating wider field adoption of EPNs by practitioners, including those operating in resource-limited settings. Framing the device as a tool that enhances the operational feasibility of biological control would strengthen the practical relevance of the study.
Comment: The manuscript currently lacks engagement with recent research on alternative devices or innovations for nematode extraction and rearing. While the novelty of the NEMA Device is clear, a brief comparison with existing or emerging tools would help position this work within the current technological landscape.
Recommendation: Consider incorporating a brief review of recent literature on alternative EPN extraction/rearing devices or systems. This would provide useful context and highlight the unique contributions of the NEMA Device.
Comment: The Discussion section does not currently outline any future research directions stemming from this study. As such, it misses an opportunity to guide follow-up work or suggest validation steps that could confirm the broader utility of the device.
Recommendation: I recommend adding a short paragraph proposing potential future studies, such as field validation of the NEMA Device in natural (non-sterile) soils, comparisons across soil types and climates, or scaling up trials for operational deployment in agricultural settings.
Conclusions
Comment: It is worth mentioning future research directions, e.g. field testing. Add a forward-looking statement about recommending further validation under field conditions and in different soil types.
“...impractical or resources are limited . ” (line 225) Comment: remove the space before the period.
Figures
Figure 2
Recommendation: Figure legends could be more precise, e.g., clarify what the “double asterisks” mean (p = 0.003 in Figure 2).
Figure 4
Comment: Inconsistency in description: The caption states that infective juveniles emerged “from G. mellonella hosts in NEMA Devices and traditional Baermann funnels”, but the figure and methods clearly refer to the use of White traps for rearing, not Baermann funnels.
Comment: In Figure 4, the y-axis lacks a clear label indicating the measurement unit or scale used to represent the total emergence of infective juveniles. This omission reduces interpretability and may lead to misinterpretation of the graphical data.
Additional comments
Please note a critical spelling error in the abstract: "Steinernema khoungi" (Abstract, line 9) The correct scientific nomenclature for this species is: Steinernema khuongi. This typographical mistake must be corrected to maintain scientific accuracy and consistency within the text. Please ensure this correction is thoroughly applied throughout the manuscript.
According to standard taxonomic practice, the name of a species should be accompanied by the name of the describing author and the year of publication at the first mention in the manuscript. (e.g., Heterorhabditis bacteriophora Poinar, 1976)
Terminology: Use full names on first appearance of terms like IJs ( lines 94, 97)
Typos and grammar: “Petri dishes containing inoculated G. mellonela.. ” ; “...from the inoculated G. mellonela (about 5-7 days) ”. Typos: "G. mellonela" instead of "G. mellonella" (lines 130, 136)
“Briefly, stock cultures are kept in tissue culture flasks at concentrations not exceeding 2500... ” (line 94) Comment: "Briefly" is generally not appropriate in scientific writing — especially in Materials and Methods sections.
“Petri dishes containing ... larvae where then added... ” (line 130) Comment: Typo: “where” should be “were”.
“...only recently emerged (less than 2wks old... ” (line 96) Comment: "2wks" is inappropriate for a scientific journal and inconsistent with formal style. It is an informal, shorthand notation.
References. Upon reviewing the references section, I observed that the scientific names of nematodes, such as "Steinernema" and "Heterorhabditis", are not consistently italicised. Comment: consistently italicize scientific names throughout.
Few minor typographical inconsistencies (e.g., spacing around parentheses): Line 116
Recommendation: While the manuscript presents results graphically, a tabular summary of the main quantitative findings would be highly valuable. This would enhance the accessibility of the results and allow for direct numerical comparison between devices.
Reviewer 2 Report
Comments and Suggestions for Authors
After reading the manuscript, I considere that although the authors present the device for a better Efficient Extraction and Rearing of Entomopathogenic Nematodes in the toitle and abstract, the efficiency extration superior to the Baerman funnel and the White trap, just between 5 to 7 % greater, and with no differences in rearing for both nematode species as it 's mentioned in page 6: Lines from 166 to 170: The NEMA Device had significantly and consistently higher recovery rates compared with the traditional Baermann funnels (t= 3.3, df = p= 0.003; Figure 2). The NEMA Device had a 5% (95% Confidence Interval [CI]: 0.1%,10%) higher recovery rate for H. bacteriophora and 7% (95% CI: 5%, 9%) higher recovery rate for S. khuongi (Figure 3)
Anyway, I think it's interesting to publish it, but I would change the title and point to, as stated in the discussion, the importance of this device as a study tool that reduces contamination and water evaporation during juvenile collection, and lays the groundwork for improving the performance of these isolation and recovery techniques, advantages in portability and versaulity..
I suggest a title like t his:
“The NEMA device improves extraction efficiency, portability, and versatility for entomopathogenic nematode applications in both the laboratory and field” or eliminate the word efficiency from the current title.
Abstract
I sugget to add the result in percentage of extrction rates for both species in the abtsract.
Methodology
It is not clear how the infective juveniles pass to the water in nema device for white trap. Water up to the capsule? Explain better
Reviewer 3 Report
Comments and Suggestions for Authors
Dear Authors,
Your idea is good, practical, but you did not describe it clearly or present it graphically precisely enough. I have given many suggestions in the report. Please consider them carefully.
Before the reparation of the manuscript, read the basic old work.
Overgaard Nielsen C.: Studies on the soil microfauna II. The soil inhabiting nematodes. Natura Jutlandica. 1949; 2: 1–131. [Google Scholar] https://www.jstor.org/stable/3564719
plus others like https://brill.com/view/journals/nema/1/3/article-p249_9.xml or https://pmc.ncbi.nlm.nih.gov/articles/PMC3578467/
Utilize their data to enhance the Introduction and Discussion.
Good luck!

All are suggested in the reviewer report.
Round 2
Reviewer 3 Report
Comments and Suggestions for Authors
Dear Authors,
It is better.
There are some suggestions:
Line 63: [7? ], ???
Lines 119-128 still are plagiarism.
Line 91: How big are soil samples?
Acknowledgments: It would be nice if you listed their affiliations.
Line 309: wrongly cited family name of the first author. It should be the author’s family name Ahlmann-Eltze. Constantin is his first name.
It would have been nice if you had the time, and use the suggested references in future work.
Comments on the Quality of English LanguageIt is better.
But, there are some suggestions:
Line 63: [7? ], ???
Lines 119-128 still are plagiarism.
Line 91: How big are soil samples?
Acknowledgment.s: It would be nice if you listed their affiliations.
Line 309: wrongly cited family name of the first author. It should be the author’s family name Ahlmann-Eltze. Constantin is his first name
